# Effects of the implementation of the HIV Treat All guidelines on key ART treatment outcomes in Namibia

**Lung Vu**[1]*, **Brady Burnett-Zieman**[1], **Lizl Stoman**[2], **Minh Luu**[3], **Johnface Mdala**[4], **Krista Granger**[1], **Steven Forsythe**[5], **Abeje Zegeye**[6], **Scott Geibel**[1]

**1** Population Council, Washington, DC, United States of America, **2** Survey Warehouse, Windhoek, Namibia, **3** Emory University, Atlanta, Georgia, United States of America, **4** IntraHealth International, Windhoek, Namibia, **5** Avenir Health, Glastonbury, CT, United States of America, **6** USAID, Windhoek, Namibia

* lvu8@worldbank.org

**Data Availability Statement:** The two data sets used to produce this manuscript are available from Harvard Dataverse (DOI: 10.7910/DVN/OZNISB and DOI: 10.7910/DVN/05DPTL).

## Abstract

### Background

This study aimed to help the Namibian government understand the impact of Treat All implementation (started on April 1, 2017) on key antiretroviral therapy (ART) outcomes, and how this transition impacts progress toward the UNAIDS's 90-90-90 HIV targets.

### Methods

We collected clinical records from two separate cohorts (before and after treat-all) of ART patients in 10 high- and medium-volume facilities in 6 northern Namibia districts. Each cohort contains 12-month data on patients' scheduled appointments and visits, health status, and viral load results. We also measured patients' wait time and perceptions of service quality using exit interviews with 300 randomly selected patients (per round). We compared ART outcomes of the two cohorts: ART initiation within 7 days from diagnosis, loss to follow-up (LTFU), missed scheduled appointments for at least 30 days, and viral suppression using unadjusted and adjusted analyses.

### Results

Among new ART clients (on ART for less than 3 months or had not yet initiated treatment as of the start date for the ART record review period), rapid ART initiation (within 7 days from diagnosis) was 5.2 times higher after Treat All than that among clients assessed before the policy took effect [AOR: 5.2 (3.8–6.9)]. However, LTFU was higher after Treat All roll-out compared to before Treat All [AOR: 1.9 (1.3–2.8)]. Established ART clients (on ART treatment for at least three months at the start date of the ART record review period) had over 3 times greater odds of achieving viral suppression after Treat All roll-out compared to established ART clients assessed before Treat All [AOR: 3.1 (1.6–5.9)].

**Funding:** This work was supported by Project SOAR (Cooperative agreement AID-OAA-A-14-00060), made possible by the generous support of the American people through the United States President's Emergency Plan for AIDS Relief (PEPFAR) and United States Agency for International Development (USAID). The contents of this paper are the sole responsibility of the authors and do not necessarily reflect the views of PEPFAR, USAID, or the United States Government. In addition, we would like to declare that the funder has never paid direct salaries to any authors, including Lizl Stoman from Survey Warehouse, for working on this manuscript.

**Competing interests:** None of the authors nor USAID alter our adherence to PLOS ONE policies on sharing data and materials.

## Conclusions and recommendations

The findings indicate positive effect of the "Treat All" implementation on ART initiation and viral suppression, and negative effect on LTFU. Additionally, by April 2018, Namibia seems to have reached the UNAIDS's 90-90-90 targets.

## Introduction

Namibia is a sparsely populated country of 2.5 million people located in sub-Saharan Africa, a region hardest hit by the HIV epidemic and is home to about 70% of all PLHIV globally (2017 data) [1]. In recent years, the region has observed a decreasing trend in both HIV infections (28% reduction compared to 2010), and deaths (44% reduction compared to 2010) [2]. A number of countries showed strong declines in new infections between 2010 and 2018, such as Rwanda, South Africa and Uganda, but new infections increased in other countries, such as Angola, Madagascar and the South Sudan [2]. Namibia, despite its remarkable achievement in providing ART to its PLHIV population, still faces one of the world's most formidable HIV epidemics, similar to its nearby Southern region countries of Lesotho, Mozambique, South Africa, Swaziland, Zambia and Zimbabwe [3]. The epidemic follows a generalized transmission pattern and is the leading cause of death in the country [4]. Recent studies indicate that 12.6% of Namibian adults (ages 15–64) are living with HIV and prevalence tends to be higher among women (15.7%) compared to men (9.3%). Based on the recent National HIV Impact Assessment, among adults living with HIV, 86% know their HIV status, 96% of those living with HIV report receiving antiretroviral therapy (ART), and 91% of those who are on ART are virally suppressed, translating to a population-level viral suppression rate of 76% [5]. This progress toward the UNAIDS 90-90-90 targets is encouraging, but despite significant health sector investments from the Government of Namibia, there is widespread social inequality. Many Namibians live highly mobile lives and often cluster in underdeveloped and remote rural communities where HIV treatment services are difficult to access [6].

In late 2015, the World Health Organization (WHO) announced new guidelines stating that anyone who has tested positive for HIV should begin ART as soon as possible [7]. Based on recent evidence that earlier ART initiation results in better clinical responses than delayed treatment, the WHO recommended the adoption of a universal Treat All or "test-and-start" approach, wherein CD4-based treatment thresholds are removed and new HIV patients are eligible to enroll in ART as soon as they test positive [8]. A body of findings from multinational trials has supported this guidance. Results from the TEMPRANO study showed that initiating ART at a CD4 count >500 cells/mm$^3$ led to significantly less severe HIV morbidity (measured as a combined outcome of death, AIDS, and severe non-AIDS diseases such as malignancies and bacterial diseases) compared to initiating ART at a CD4 count < = 500 cells/mm$^3$ [9]. The START Trial showed that the immediate initiation of antiretroviral therapy among previously untreated adults living with HIV with a CD4 count >500 cells/mm$^3$ was superior to initiation that was delayed until the CD4 count declined to 350 cells/mm$^3$, based on a composite outcome including serious AIDS-related and non-AIDS related events [10]. A recent meta-analysis study further validated the effectiveness (reduced mortality risk and risk for AIDS) of early initiation of ART as defined in the current WHO guidelines [11].

Prior to adopting the "Treat All" guidelines, Namibia followed the 2014 WHO guidelines for ART eligibility, under which children under 15 years of age, pregnant women, HIV-discordant couples, and HIV-positive patients who either had a CD4 count ≤500 cells/mm3 or had been classified in WHO-stages 3 or 4 were eligible for ART treatment [7, 12]. As the 2015

guidelines were introduced, they did not quantify a recommended timeframe from HIV diagnosis to ART initiation nor officially adopted the guidance, rather they suggested that efforts be made to reduce time between HIV diagnosis and ART initiation for those who are ready to start treatment [7]. In 2017, WHO updated the guidelines, recommending that rapid ART initiation—defined as initiating ART and receiving the first dose of antiretroviral medication within seven days of diagnosis—should be offered to people living with HIV following a confirmed diagnosis and clinical assessment [13]. Further, ART initiation is recommended on the same day as HIV diagnosis if a person has received pre-treatment counseling and feels ready to begin treatment [13]. Following the addition of the new rapid initiation recommendations into WHO's Treat All guidelines, the Government of Namibia adopted the full Treat All guidelines as national policy, effective from April 2017 [14]. Namibia, together with Botswana, eSwatini, South Africa and Malawi, were the first in sub-Saharan Africa rolling out the Treat All guidelines nationwide.

This study was designed to aid the Namibian government in understanding the effects of this national implementation of Treat All guidelines on key ART outcomes at selected sites, and how this transition may affect progress toward 90-90-90 targets. Through these initiatives, the Government of Namibia has demonstrated a commitment to expanding access to treatment and is moving towards to achieving 90-90-90 targets [14]. As Namibia began the piloting and phased rollout of Treat All, there have appeared many unanswered questions about its broader effects—not only on client-level treatment outcomes, but also on the health system level, including demands on limited human and financial resources. In this paper, we will report findings from our assessment of Namibia's Treat All implementation, including its effects on key HIV indicators: ART initiation, retention in ART, viral suppression, and service quality. We aim to measure the effect as well as lessons learned after one year implementing Treat All and inform scale up of this approach in Namibia and elsewhere.

## Materials and methods

### Study site and design

This is implementation science study uses retrospective, observational data extracted from longitudinal clinical health record data, and data collected through exit interviews from a random sample of HIV care and treatment clients at health facilities in Namibia at 2 separate timepoints. The site selection was determined based on careful and thorough discussion with the Namibian government, USAID and key implementing partner (IntraHealth). We used routine ART service records to assess monthly client volumes and geographic representation. To ensure data accessibility and reliability of health records, we included only sites that had received data quality improvement support from IntraHealth International through the United States Agency for International Development Clinical Services Technical Assistance Project (UTAP). Because Namibia has a small population (2.5 million), we purposively selected the 10 largest health facilities, covering both urban and rural settings, in order to capture as many ART clients who initiated ART during the data review periods as possible. These 10 sites accounted for 75% of PLHIV in the country and were all located in Northern Namibia. Six district-level hospitals (centralized hospitals) and four rural health centers (decentralized/ community-based facilities) were ultimately selected.

**Health record extraction.** We conducted two rounds of record reviews to collect retrospective data on two cohorts of clients who accessed ART services at the selected facilities during either the year before (Round 1: 1 April 2016–31 March 2017) or the year after (Round 2: 1 April 2017–31 March 2018) the national adoption of Treat All. Prior to each round of data collection, we obtained a sampling frame of adult ART clients who were either engaged in ART

services as of the 1 April start date for the corresponding time period. We also captured data on adult ART clients or who had received a confirmed HIV-positive diagnosis after 1 April of the corresponding year and subsequently initiated ART. The list of facilities can be found elsewhere [15]. Baseline retrospective data collection took place between June—August 2017; endline occurred between June—August 2018. Clients who were either lost to follow-up (i.e., those who had missed their most recent appointment by over 90 days and had not returned to the clinic) or were under age 18 at the beginning of the respective review period were excluded. At site level, clients were selected randomly from the ten selected facilities in proportion to each facility's ART population. Data collectors captured all information from the client care booklets, including client demographics; dates of confirmatory HIV diagnosis and subsequent ART initiation; clinical visit dates; drug dispensing data; dates of follow-up appointments; and viral load testing dates and results.

**Client exit interview.** We conducted exit surveys at each health facility to compare client satisfaction during early stages of Treat All rollout (June to July 2017) and after rollout (June to July 2018). To capture a diverse sample of ART patients, we conducted the exit surveys on five consecutive days within one week, and interview timing was spaced to occur throughout the day. Interviewers approached potential clients as they left their final consultation of the day. Across the 10 study sites, we interviewed 310 ART clients close to baseline (2 months after TA policy was announced by the Namibian government) and 286 clients at endline (about a year after TA). We measured clients' self-reported wait times and perceptions of service quality. Wait time was self-reported, which includes: 1) wait time from arrival at facility to meeting the intake nurse; 2) wait time to see a clinical HIV provider; 3) wait time at the laboratory; and 4) wait time to get a refill (at pharmacy). Perceptions of services quality were adapted from the patient satisfaction questionnaire short form (PSQ-18 [16]) assessing provider attitudes, care accessibility, and overall client satisfaction. Sample questionnaire items include "doctors/ nurses sometimes ignore what I tell them" or "when I go for medical care, they are careful to check everything when treating me". HIV patients were asked to rate these statements using 5 options ranging from "strongly disagree" to "strongly agree". For the purpose of this paper, we reported the percentage who agreed or strongly agreed to these statements.

**Data management.** We used tablet-based data collection tools developed using the SurveyCTO electronic data collection platform. Skip patterns and data quality checks were built in the questionnaires. All research assistants/ interviewers went through a 3-day intensive data collection and ethical training and a 2-day field-testing and practice of the data capture tools.

**Table 1. Client types and key outcome measures.**

| Outcome | Definition |
|---|---|
| Loss to follow-up | Proportion of newly initiating ART clients who missed scheduled appointment by >90 days |
| Viral suppression | Proportion of randomly sampled ART client records with documented viral load test results during review period (viral load <1,000 copies/ml) |
| Rapid ART initiation | Initiated ART within seven days of diagnosis. |
| New ART clients | ART clients who had been on ART for less than 3 months, or had not yet initiated treatment as of the start date for the review period |
| Established ART clients | ART clients who had been receiving ART treatment for at least three months, and were not lost to follow-up, as of the start date of the review period. |
| Round 1 data | Before Treat All, containing data of ART patients from April 1, 2016 to April 1, 2017 |
| Round 2 data | After Treat All, containing data of ART patients from April 1, 2017 to April 1, 2018 |

### Definitions of key outcomes and client types

Definitions of key outcomes and client types are described in Table 1.

### Data analysis

We compared treatment outcomes and client satisfaction before and after Treat All implementation and between district hospitals and rural health centers. We conducted bivariate analyses including chi-squared and Fisher's exact tests to assess differences in key outcomes between two time points and the two service modalities. T-tests were used to assess differences in means; non-parametric tests were used to assess differences in medians (test of equality). Finally, we used multiple logistic regression to examine four key outcomes: ART initiation, one-year retention, missed an appointment for more than 30 days, and viral suppression. These models adjusted for relevant socio-economic factors, including facility, gender, age, and marital status. All analyses were done using Stata version 15 (StataCorp., College Station, TX).

### Ethical approval

The study was approved by the Institutional Review Board of the Population Council and the Biomedical Research Ethics Committee of the Ministry of Health and Social Services in Namibia. Both routine and exit interview data contained no personal information (such as name, date of birth, address, phone) of ART clients. All exit interview clients provided written informed consent.

## Results

### Demographic characteristics of the study populations

We reviewed the health records of 2,293 clients— 1,179 clients at Round 1 (before Treat All) and 1,114 clients at Round 2 (after Treat All). As shown in Table 2, two-thirds of all clients were female; median age was 40 at Round 1 and 39 at Round 2. There were significant differences in marital status between Round 1 and Round 2 samples with a higher proportion reported never married in Round 2 (71% vs. 65%; p = 0.001). A larger proportion of clients at Round 2 attended district hospitals compared to Round 1 (80% vs. 70%, respectively, p<0.001). Clients' time on ART differed significantly between Rounds 1 and 2 (p = 0.001), and at Round 2, clients were more likely to have started ART at a lower clinical stage (74% started at stage 1, less severe) compared to clients at Round 1 (61% started ART at stage 1; p < 0.001) and a smaller proportion of clients were established (or current) ART clients (p < 0.001).

### Factors associated with key ART outcomes

Results in Table 3 (bivariate relationship) suggest that the "after Treat All" period, younger age, being single, clinical stage 1, and client type were associated with rapid ART initiation (beginning treatment within seven days of diagnosis). Additionally, loss to follow-up (LTFU) was associated with the after Treat All period, younger age, being male, and recent ART initiation. Table 2 results also indicate "missing a routine care appointment" for more than 30 days was associated with client type, while achieving viral suppression was associated with the after Treat All period, older age, being female, and clinical stage 4.

Multiple logistic regression results comparing key indicators before and after Treat All roll-out are presented in Table 4, stratified by new ART clients and established ART clients. Among new ART clients, after adjusting for age, sex, marital status, facility type, and clinical stage, rapid ART initiation was 5.2 times higher after Treat All than that among clients reviewed before the policy took effect [AOR: 5.2 (3.8–6.9)]. However, LTFU was also higher after Treat All roll-out compared to before Treat All [AOR: 1.9 (1.3–2.8)]. There were no

**Table 2. Sociodemographic characteristics of the pre and post Treat All samples.**

| | Before Treat All | After Treat All | Total | Chi2 |
|---|---|---|---|---|
| | % (N) | % (N) | % (N) | (p-value) |
| **Sex** | | | | |
| Male | 33.3 (393) | 34.0 (379) | 33.7 (772) | *0.12 (0.73)* |
| Female | 66.7 (786) | 66.0 (735) | 66.3 (1,521) | |
| **Median age, years (IQR)** | 40 (33–47) | 39 (31–48) | 39 (32–47) | *0.678 (0.410)* |
| **Marital status**** | | | | |
| Never married | 64.5 (734) | 70.9 (759) | 67.6 (1,493) | *13.18 (0.001)* |
| Married | 30.5 (347) | 26.2 (280) | 28.4 (627) | |
| Divorced | 5.0 (57) | 2.9 (31) | 4.0 (88) | |
| **Facility** *** | | | | |
| District hospital | 70.1 (826) | 79.9 (890) | 74.8 (1,716) | *29.41 (0.001)* |
| Rural health center | 29.9 (353) | 20.1 (224) | 25.2 (577) | |
| **Client type**** | | | | |
| New ART | 46.4 (484) | 55.6 (695) | 51.4 (1,179) | 19.62 (0.000) |
| Established ART | 53.6 (560) | 44.4 (554) | 48.6 (1,114) | |
| **Time on ART (years)** *** | | | | |
| Mean (std) | 4.1 (3.7) | 3.5 (3.8) | 3.8 (3.7) | *3.5 (<0.001)* |
| Median (IQR) | 2.6 (1–6.8) | 1.2 (0.9–5.8) | 2.9 (1.0–6.4) | *21.9 (<0.001)* |
| **Clinical stage at initiation** *** | | | | |
| Stage I | 61.4 (662) | 74.3 (750) | 67.6 (1,412) | *49.78 (<0.001)* |
| Stage II | 18.8 (203) | 15.0 (151) | 16.9 (354) | |
| Stage III | 17.9 (193) | 8.8 (89) | 13.5 (282) | |
| Stage IV | 1.9 (21) | 2.0 (20) | 2.0 (41) | |

**Note:** t-test (mean); nonparametric test (median test of equality).

statistically significant differences with regard to either missing a routine HIV care appointment or viral suppression when comparing the before and after Treat All periods.

Established ART clients had over 3 times greater odds of achieving viral suppression after Treat All roll-out compared to established ART clients assessed before Treat All [AOR: 3.1 (1.6–5.9)]. There was no statistically significant difference among established ART clients between the before and after Treat All periods with regard to rapid initiation, loss to follow-up, or missed scheduled appointment (Table 4).

## Wait time and perception of service quality

Table 5 summarizes data on clients' wait time and perception of service quality. On average, participants spent nearly 2 hours waiting at the facility reception, between reception and doctor, doctor and pharmacy or lab during the first round of client exit surveys. By the second round, total wait times decreased to 85 minutes ($p < 0.0001$). These overall decreases were largely driven by district hospitals, where total wait times decreased from 120 to 80 minutes ($p<0.0001$). The greatest reductions in wait times were at reception (69 to 42 minutes; $p<0.0001$), pre-consultation (33 to 21 minutes; $p<0.0001$) and, to a lesser degree, pharmacies (19 to 16 minutes, $p = 0.036$). Mean time spent traveling decreased among both district hospitals and rural health centers between Rounds 1 and 2 (70 minutes vs. 51 minutes, respectively, $p<0.0001$). After Treat All, some clients perceived a higher quality of personal interactions with care providers compared to before Treat All. Most notably, 98% (98% after Treat All vs.

**Table 3. Factors associated with key ART outcomes (bivariate level).**

| | Rapid ART initiation ($\leq$ 7 days) | Lost to follow-up | Missed appointment ($\geq$30 days)[†] | Achieved viral suppression[‡] |
|---|---|---|---|---|
| | % (N) | % (N) | % (N) | % (N) |
| **Time** | | | | |
| Before Treat All | 11.3 (133)*** | 9.4 (111)*** | 15.0 (160) | 91.6 (565)** |
| After Treat All | 30.0 (334) | 14.2 (158) | 14.9 (142) | 96.2 (652) |
| **Age** | | | | |
| <30 | 38.1 (162)*** | 18.1 (77)*** | 16.7 (58) | 85.0 (164)*** |
| $\geq$30 | 16.3 (305) | 10.3 (192) | 14.6 (244) | 95.6 (1,053) |
| **Sex** | | | | |
| Male | 18.8 (145) | 13.7 (106)* | 313.7 (91) | 91.9 (388)* |
| Female | 21.2 (322) | 10.7 (163) | 15.5 (211) | 95.0 (829) |
| **Facility type** | | | | |
| District-level hospital | 21.2 (364) | 12.4 (213) | 14.6 (219) | 94.1 (934) |
| Rural health clinics | 17.8 (103) | 9.7 (56) | 15.9 (83) | 93.4 (283) |
| **Marital status** | | | | |
| Single/never married | 22.6 (338)** | 12.46 (186) | 13.5 (177) | 93.3 (787) |
| Married/cohabitating | 17.1 (107) | 9.9 (62) | 16.5 (93) | 94.1 (328) |
| Divorced/separated widowed | 14.8 (75) | 11.4 (10) | 15.4 (12) | 95.7 (45) |
| **Clinical stage at time of initiation** | | | | |
| Stage I | 23.2 (328)*** | 11.8 (167) | 13.9 (173) | 95.2 (738)* |
| Stage II | 13.3 (47) | 10.7 (38) | 15.8 (50) | 91.6 (196) |
| Stage III | 10.3 (29) | 12.1 (34) | 20.2 (50) | 90.4 (132) |
| Stage IV | 19.5 (8) | 17.1 (7) | 14.7 (5) | 100.0 (22) |
| **Client type** | | | | |
| New ART | 37.1 (387)*** | 14.0 (1146)** | 14.6 (131) | 95.1 (463) |
| Established ART | 6.4 (80) | 9.9(1,249) | 15.2 (171) | 93.4 (777) |

**Note:** (*) p-value < 0.05; (**) p-value < 0.01; (***) p-value < 0.001 (†) n = 2,024 clients who were not lost to follow-up; (‡) n = 1,295 clients who received a viral load test.

95% before Treat All; p = 0.037) agreed that they were treated respectfully, and fewer (3.9% after Treat All vs. 17.8% before Treat All; p = 0.001) felt that doctors/ nurses sometimes ignored what clients told them. Very few clients in either round agreed that they sometimes wonder whether their provider's diagnoses are correct (6% vs 8.4%, p = 0.25). However, a

**Table 4. Comparisons of key treatment outcomes: Multivariate level.**

| | Newly initiated ART clients | Established ART clients |
|---|---|---|
| | N = 1,044 | N = 1,249 |
| | After Treat All | After Treat All |
| *Outcomes* | *Adjusted odds ratio (95% CI)* | *Adjusted odds ratio (95% CI)* |
| **Rapid ART initiation within 7 days** | 5.2 (3.8–6.9)*** | 1.2 (0.76–2.0) |
| **Loss to follow-up** | 1.9 (1.3–2.8)*** | 1.1 (0.8–1.6) |
| **Missed scheduled appointment >+30 days** | 1.2 (0.8–1.8) | 0.9 (0.6–1.2) |
| **Achieved viral suppression** | 1.3 (0.5–3.1) | 3.1 (1.6–5.9)*** |

**Notes:** 1) Variables adjusted for in the multivariate analysis: age, sex, marital status, facility type, clinical stage; 2) AOR: comparing after Treat All to before Treat All.

**Table 5. Wait time and perception of service quality (proportion who agreed/strongly agreed with each statement).**

| | Before Treat All | After Treat All | Chi$^2$ or t-test |
|---|---|---|---|
| | % (n = 310) | % (n = 285) | (p-value) |
| Mean total wait time in minutes (std) | 118 (84.6) | 85 (58.1) | *5.5 (<0.001)* |
| Median wait time in minutes (IQR) | 100 (55–170) | 75 (42–108) | *27.4 (<0.001)* |
| My doctors/ nurses treat me in a very respectful manner. | 95.1 | 98.3 | *4.4 (0.037)* |
| Doctors/ nurses sometimes ignore what I tell them. | 17.7 | 3.9 | *29.0 (<0.001)* |
| Sometimes doctors/ nurses make me wonder if their diagnosis is correct. | 8.4 | 6.0 | 1.3 (0.254) |
| My HIV care doctors/ nurses usually spend plenty of time with me. | 87.7 | 57.2 | *70.5 (0.001)* |
| I have easy access to the medical specialists I need. | 92.0 | 81.8 | *13.7 (<0.001)* |
| I am able to access medical care whenever I need it. | 94.8 | 87.0 | *11.2 (0.001)* |
| When I go for medical care, they are careful to check everything when treating me. | 95.5 | 91.6 | 3. 8 (0.052) |

**Note**: t-test (mean); nonparametric test (median).

lower percentage of clients agreed that their doctors/ nurses spent enough time with them after Treat All, compared to before (57% vs. 88%, p = 0.001). Furthermore, there was a significant reduction after Treat All in the proportion of clients who stated that they sometimes had easy access to the medical specialists (81.8% vs 92.0%; p<0.001), or to the medical care they need (87.0% vs 94.8%; p = 0.001), and marginally fewer clients said that clinicians were careful to check everything while treating them (91.6% vs. 95.5%; p = 0.052) (Table 5).

## Discussion

These data show that the facilities surveyed reached two critical milestones: over 90% of ART patients were retained in care after 12 months and 90% of clients who received a viral load test had achieved viral suppression. Our findings are consistent with the Namibia National HIV Impact Assessment (2018) [5, 17], which indicates that Namibia is the first country to have three quarters of its HIV-positive population virally suppressed, a good indicator of a successful HIV treatment program. Consistent with findings from Eswatini, time to initiation under Treat All decreased considerably compared to the standard of care [18]. Furthermore, the implementation of the Treat All guidelines has had positive effects on health outcomes. One year after the implementation of Treat All, patients were more likely to achieve viral suppression. As anticipated based on evidence from a recent systematic review, it appears that the Treat All guidelines, along with the decentralization of clinical services in Namibia, are contributing to better health outcomes for people living with HIV [19, 20].

However, we also found that Treat All may negatively affect retention among newly initiated ART clients. This finding is aligned with several recent completed studies investigating the impact of the test and start approach. These studies found that early ART initiation showed a strong impact on linkages to care, but LTFU increased during the first two to three months after ART initiation [21–23]. There are ART clients who were not psychologically ready to accept their HIV-positive diagnosis and ART treatment, as well as low awareness, and therefore low acceptability of the Treat All guidelines [24]. It is critical that readiness assessments of clients and initial counseling messages properly address the support needs of new ART clients

and ensure they are equipped to adhere to lifelong ART immediately after their HIV-positive diagnosis. Additionally, raising awareness of the Treat All guidelines through health communication campaigns is necessary. Although HIV providers strictly follow the national treatment guidelines and provided adherence counseling and scheduled the first visit one month after ART initiation, additional follow-ups via phone or home visit may be needed for patients initiating ART on the day of diagnosis to provide timely support and track movement of these clients along the care continuum.

Median wait time decreased significantly (100 minutes vs. 75 minutes; p<0.01). The decentralization of ART services and implementation of community-level ART services have improved and are expected to continue to facilitate better ART adherence among ART clients because these strategies can significantly reduce travel and wait time due to reduced overcrowding at centralized hospitals. Specifically, since 2016, stable ART clients from district-level hospitals were allowed to transfer to smaller community health centers to expand ART access in rural areas. In addition, Community-based ART nurses and pharmacist/health assistants from central hospitals coordinate with health extension workers and expert patients (long term patients) to deliver ART services and refills at community-based drug distribution points every 3 months. Bringing services closer to home can reduce travel associated costs, resulting in fewer missed appointments and LTFU. These effects have been found in many other settings and patient outcomes have been found to be similar to that of those who only receive care at facilities [25–27].

Despite improvements in wait times, results of patient satisfaction surveys were mixed. There was greater satisfaction reported on items representing individual interactions with practitioners, however, clients also reported reduced ability to access the services they need, or to spend sufficient time with their providers. This is consistent with our previous publication that shows a significant reduction of clinical time between before treat all and after treat all [15]. This may be due to increased demands on providers time, or the use of community-based ART delivery strategies. There is a need, however, to carefully monitor these models of care. In other countries, for example, weakness in drug supply chains have challenged the success of HIV care outside of district hospitals [26]. Availability of and support for trained community health workers or patient peers to enable the success of models for delivering ART in communities has also proven a challenge in other countries and needs to be carefully provided for and monitored in Namibia [28].

## Limitations

Our study has several limitations. First, in order to infer conclusions about the effect of Treat All on key outcomes, an ideal study design would have involved at least a two-arm comparison between Treat All and non-Treat All. However, our study aimed to capture the real-life implementation of Treat All under the pressure of time and limited budget. For this reason, our approach using existing routine data and client exit surveys seemed well suited. In addition, due to the nationwide rollout of the Treat All implementation, it was not feasible to establish a rigorous comparison and it would have been unethical to have a control population that did not receive Treat All. Aside from the design limitations, the data were collected retrospectively and thus subjected to errors and various data quality issues. These are common limitations of routine program data, especially data from small health facilities in limited resource settings in Northern Namibia. It is important to note that longitudinal data was not feasible to collect due to challenges in recruiting enough newly positive clients (Namibia has a small population size) at IntraHealth supported sites within a limited timeframe. Lastly, our client exit survey was likely subjected to social desirability bias and recall bias, especially when asking about clients'

travel and wait time and when interviews were being conducted in a limited-space health facility.

## Conclusion and recommendations

As the WHO's Treat All guidelines are initiating implementation in countries worldwide, the current roll out of Treat All in Namibia can help provide insight into how these guidelines might work within the context of existing and other novel differentiated care models. This study demonstrates that, the implementation of the Treat All guidelines has made a positive impact on several key treatment outcomes. In particular, one year after the national rollout of the guidelines (by April 2018), Namibia seems to have reached the last two of the UNAIDS's 90-90-90 targets. Second, while Treat All may have expanded patient volume, service decentralization was likely attributed to reduced patient wait time while maintaining high service quality. However, greater LTFU among newly initiated ART clients is a potential issue that may result from the rollout of Treat All. We note, however, that this was a relatively short-term study and therefore, monitoring key ART outcomes, including service quality is important. Particularly, we recommend careful monitoring of patients initiating ART on the day of HIV diagnosis as a strategy to reduce LTFU and to reach the next-level "95-95-95" targets (UNAIDS targets by 2030) [26, 29]. Lastly, the study has demonstrated the feasibility of using existing data to measure key clinical outcomes while strengthening local organizations on collecting and using routine data. Collecting, analyzing, and utilizing existing data on key clinical outcomes and ART-related cost works best with high-quality and complete data, which can be a challenge logistically in many settings [30], but proved to be effective in Namibia.

## Acknowledgments

We would like to thank the data collection team from Survey Warehouse who worked tirelessly to ensure the success of this study. We especially thank our implementing partner—Intra Health International (Namibia) that gave us access to critical ART service data and participated in the planning, data collection and dissemination of the findings. Last but not least, we sincerely thank the Ministry of Health and Social Services that provided leadership and support throughout the study.

## Author Contributions

**Conceptualization:** Lung Vu, Brady Burnett-Zieman, Minh Luu, Abeje Zegeye, Scott Geibel.

**Data curation:** Brady Burnett-Zieman, Minh Luu, Johnface Mdala.

**Formal analysis:** Lung Vu, Brady Burnett-Zieman.

**Funding acquisition:** Lung Vu, Scott Geibel.

**Investigation:** Lung Vu, Lizl Stoman, Johnface Mdala.

**Methodology:** Steven Forsythe.

**Project administration:** Brady Burnett-Zieman, Lizl Stoman.

**Supervision:** Johnface Mdala.

**Validation:** Lung Vu.

**Writing – original draft:** Lung Vu.

**Writing – review & editing:** Brady Burnett-Zieman, Lizl Stoman, Minh Luu, Johnface Mdala, Krista Granger, Steven Forsythe, Abeje Zegeye, Scott Geibel.

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
