## [Decision Letter · Decision Letter 0]

26 Jun 2020

PONE-D-20-12826

Effects of the implementation of the HIV Treat All guidelines on key ART treatment outcomes in Namibia

PLOS ONE

Dear Dr. Vu,

Thank you for submitting your manuscript to PLOS ONE. After careful consideration, we feel that it has merit but does not fully meet PLOS ONE’s publication criteria as it currently stands. Therefore, we invite you to submit a revised version of the manuscript that addresses the points raised during the review process.

We look forward to receiving your revised manuscript.

Kind regards,

Matthew Quaife

Academic Editor

PLOS ONE

Journal Requirements:

2. In ethics statement in the manuscript and in the online submission form, please provide additional information about the patient records used in your retrospective study. Specifically, please ensure that you have discussed whether all data were fully anonymized before you accessed them and/or whether the IRB or ethics committee waived the requirement for informed consent. If patients provided informed written consent to have data from their medical records used in research, please include this information.

"This work was supported by Project SOAR (Cooperative agreement AID-OAA-A-14-00060), made possible by the generous support of the American people through the United States President’s Emergency Plan for AIDS Relief (PEPFAR) and United States Agency for International Development (USAID). The contents of this paper are the sole responsibility of the authors and do not necessarily reflect the views of PEPFAR, USAID, or the United States Government. "

We note that one or more of the authors are employed by a commercial company: Survey Warehouse.

3.1. Please provide an amended Funding Statement declaring this commercial affiliation, as well as a statement regarding the Role of Funders in your study. If the funding organization did not play a role in the study design, data collection and analysis, decision to publish, or preparation of the manuscript and only provided financial support in the form of authors' salaries and/or research materials, please review your statements relating to the author contributions, and ensure you have specifically and accurately indicated the role(s) that these authors had in your study. You can update author roles in the Author Contributions section of the online submission form.

3.2. Please also provide an updated Competing Interests Statement declaring this commercial affiliation along with any other relevant declarations relating to employment, consultancy, patents, products in development, or marketed products, etc. 

5. We note you have included a table to which you do not refer in the text of your manuscript. Please ensure that you refer to Table 4 in your text; if accepted, production will need this reference to link the reader to the Table.

Reviewers' comments:

Reviewer's Responses to Questions

**Comments to the Author**

1. Is the manuscript technically sound, and do the data support the conclusions?

Reviewer #1: Yes

Reviewer #2: Yes

2. Has the statistical analysis been performed appropriately and rigorously? 

Reviewer #1: No

Reviewer #2: Yes

3. Have the authors made all data underlying the findings in their manuscript fully available?

Reviewer #1: Yes

Reviewer #2: No

4. Is the manuscript presented in an intelligible fashion and written in standard English?

Reviewer #1: Yes

Reviewer #2: Yes

5. Review Comments to the Author

Reviewer #1: Reviewer Comments for Manuscript PONE-D-20-12826

General Comments:

The paper addresses and important topic and gives data on what is happening since Test and Treat was implemented. The flow is okay but methods need more detail on what was done and the different outcomes, hoe they were defined and how they were measured.

Introduction

1. Line 47, may give a clearer picture of Namibia if you give some data on the state of the HIV epidemic in sub-Saharan Africa (SSA) and how the situation in Namibia compares with SSA.

2. Lines 58, 79 introduce times for WHO updated guidelines. Did health facilities go on to implement the 2015 guidance on treating as soon as possible after diagnosis or was nothing done until the 2017 guidelines which gave a more specific time of 7 days. We are missing some information for the period after the 2015 guidelines.

Methods:

The methods section needs more clarity on what exactly is being measured as the outcome(s).

1. Study design: You used mixed methods, a retrospective record review and exit interviews. Discuss your study methods again with the team.

2. Study sites: You mention northern Namibia and yet this is data to inform the government. Where there no UTAP sites in other regions? Is the population distributed largely in the north and therefore representative of the country? Please justify the choice of only the northern region for data meant to inform the country.

3. Line 113 &114: Add the year to the 1st April dates make it clearer which period you are referring to.

4. Line 116 & 117: LTFU is one of the study outcomes, why did you exclude those who were lost after 90 days. Give a justification for this exclusion otherwise it creates bias in estimating the study outcome.

5. Line 132: How did the clients record time that they self-reported? Was there a reliable measure or was it just their perception of whether they waited too long or not?

6. Lines 133-138: These give an idea of how satisfaction with service quality was measured. Similarly give a definition for waiting time e.g. cumulative time waiting at certain selected service points like counselling room, lab to give a sample for VL, pharmacy etc. or is it time from arrival at the facility to exit regardless of what happens during that time including clients staying around to wait for their friends?

7. Line 140: “New ART clients” and established ART clients are not study outcomes, remove them from the table and replace with “ART initiation” and missed appointment for > 30 days”.

8. Line 145, I don’t think you are able to assess changes over time i.e. throughout the 12 month follow up. You have only 2 measurements, baseline and end line measurements.

9. Line 146: Mention the non-parametric test you used.

10. Data management: Give details of data collection tools used (paper vs electronic), the software used for data entry and how validation and query resolution were done.

Results

1. Lines 158 and 159: The concept of rounds 1 and 2 are introduced for the first time here. These are implied in the section of record extraction (line 108). Go back to methods section and clarify if clients included in round 1 were not included in round 2 and how this was mitigated or was it the same clients having a before and after measurement so that you compared study outcomes at two time points rather than comparing individuals included in the two rounds. Make this clearer to the reader in methods.

2. Lines 164, 165: The text does not match the data, please correct the statement to show that a higher proportion of clients started ART at stage 2 compared to stage 1 (74% vs 61%; p=0.001).

3. Lines 166, 167: It is not clear which round had the smaller proportion of established clients!

4. Table 1: You have used “Before treat All” and “After treat All”. How do these differ from round 1 and round 2? If these are different, please go back to the methods and make it clear what each refers to in terms of periods of the year/ which year. If there are the same, choose one term for consistency either use “rounds” or “before/ after treat All”. It gets confusing as one reads the paper.

5. Lines 175, 176: This is the definition for ART initiation that needs to be included in the study outcomes section of methods so that you do not have to repeat it in subsequent sections of the paper.

6. Lines 207-211: It is not clear in methods how clients assessed this time. Clarify this in the methods.

7. Line 222: What does marginally fewer mean?

8. Lines 217-218: Making a diagnosis is something one gets by training and experience. How are the clients able to judge that the diagnosis is not right? It could be implied from the other assessments e.g. health workers do not listen enough, or do not check everything while treating them. These other assessments of patient satisfaction are ok, the one for whether diagnosis is correct should be left out since the p-value does not even show statistical significance.

Discussion

1. Line 237: Post implementation is another term added. Be consistent with terms used in the text and tables.

2. Lines 237-239: These are results, take them to the results section and use the discussion section to “discuss the results”.

3. Lines 248-250: Include the procedures that were done in the UTAP facilities during the time of ART initiation and the days/ weeks after. Did they just let clients go or scheduled follow up counselling sessions e.g. for those who initiated same day treatment. Bringing this information out helps discussion of the results on retention.

4. Line 263: Table 5 should be in the results section.

Study Limitations

1. Lines 283-285: ART facilities in most countries have data during the time when ART was initiated using CD4 counts. Were these records not available in Namibia from 2010-2014 so that this could be a comparison group?

2. Line 288: Recall bias may not arise during records review but more the issues to do with program data not suiting the study purposes so that you end up with missing data on key variables. Also transcribing data from program data bases or tools to the study tools may have led to errors in data collection.

3. What proportion of data was missing? Some proportions may be too small that they do not have an effect, state the proportion that was missing only if it is large enough to reduce your study power to detect some associations.

Conclusion

1. Lines 304-305: You could take this to limitations if short follow up time had an effect on your study results. It is not a conclusion.

2. Focus your conclusion on results that you have on ART initiation and high LTFU after implementing new guidelines. Recommendations should come from these.

3. Lines 308-312: Your study did was not done to demonstrated the feasibility of using existing data to measure key clinical outcomes while strengthening local organizations on collecting and using routine data. It doesn’t seem to fit well in the conclusion section.

Additional reading

You can read through this article that focused on LTFU during “test and treat”. Definition used for T&T was before the 2017 guidelines.

Reviewer #2: I enjoyed reviewing this manuscript and learning more about the implementation of Test and Start in Namibia. The successes offer important insights as other Sub-Saharan African countries seek to scale-up treatment as prevention. The paper also offers methodological insights, through the use of retrospective data, as a randomized design with a control arm is not ethically or practically feasible. I just have a few minor questions and suggestions for the authors.

- In both the introduction and the discussion, I think the authors could say a bit more about how the roll-out of test and start has gone in other Sub-Saharan African (SSA) countries (particularly around the impacts on the health system). Brault, et al. 2020 in Current HIV/AIDS Reports provides a number of references that might be salient. Publications from the MaxART study in Swaziland/eSwatini describe some similar challenges with implementation of test and start in public facilities.

- More information is needed on the site selection—why only sites in northern Namibia? Also, what was the total number of eligible sites?

- It would also be helpful to understand a bit more about the communities/catchment areas of the health facilities. For instance, in and out-migration has been described as a significant factor influencing LTFU and attainment of 90-90-90 goals (here again, see either Brault, et al. 2019 in JAIDS or Brault, et al. 2020). Is this an issue in these areas?

- Testing data (the first 90) is not mentioned at all in the paper. Is testing not an issue in Namibia? If so, this should be explained. If Namibia (like many other SSA countries) continues to struggle with adequately identifying and testing difficult-to-reach populations, then why was this not explored in the present study?

- The placement of Namibia’s service decentralization timeline in the discussion feels a little out of place. I think this information should be placed in either the introduction or early in the methods.

- The authors mention the dual challenges of LTFU and lack of facility staff and/or community health workers who can assess client readiness and support new clients. Do they have any recommendations or are there any plans in Namibia to address these issues?

- The authors acknowledge that key limitations of their study are related to time and budget constraints that in turn influenced the design. This has been a challenge for many of the other studies of test and start. Do the authors recommend using a retrospective design to combat these issues, or do they have any recommendations for how to go about this differently in the future?

6. PLOS authors have the option to publish the peer review history of their article (what does this mean?). If published, this will include your full peer review and any attached files.

Reviewer #1: No

Reviewer #2: No

---

## [Author Response · Author response to Decision Letter 0]

21 Sep 2020

Please see attached the response memo.

---

## [Decision Letter · Decision Letter 1]

5 Nov 2020

PONE-D-20-12826R1

Effects of the implementation of the HIV Treat All guidelines on key ART treatment outcomes in Namibia

PLOS ONE

Dear Dr. Vu,

Thank you for submitting your manuscript to PLOS ONE. After careful consideration, we feel that it has merit but does not fully meet PLOS ONE’s publication criteria as it currently stands. Therefore, we invite you to submit a revised version of the manuscript that addresses the points raised during the review process.

We look forward to receiving your revised manuscript.

Kind regards,

Matthew Quaife

Academic Editor

PLOS ONE

Additional Editor Comments (if provided):

Thank you for the revisions. You will see a few minor comments that we would appreciate your responses to, then we will be happy to accept.

Reviewers' comments:

Reviewer's Responses to Questions

**Comments to the Author**

1. If the authors have adequately addressed your comments raised in a previous round of review and you feel that this manuscript is now acceptable for publication, you may indicate that here to bypass the “Comments to the Author” section, enter your conflict of interest statement in the “Confidential to Editor” section, and submit your "Accept" recommendation.

Reviewer #1: (No Response)

Reviewer #2: (No Response)

2. Is the manuscript technically sound, and do the data support the conclusions?

Reviewer #1: Yes

Reviewer #2: Yes

3. Has the statistical analysis been performed appropriately and rigorously? 

Reviewer #1: Yes

Reviewer #2: Yes

4. Have the authors made all data underlying the findings in their manuscript fully available?

Reviewer #1: Yes

Reviewer #2: Yes

5. Is the manuscript presented in an intelligible fashion and written in standard English?

Reviewer #1: Yes

Reviewer #2: Yes

6. Review Comments to the Author

Reviewer #1: General Comments:

Reviewer comments have been addressed, the paper is accepted pending a minor correction as mentioned below.

Minor comment:

Lines 129 & 130: Add the year to the 1st April dates to make it clearer which period you are referring to.

Reviewer #2: The authors have mostly addressed the comments in the previous manuscript. However, they stated that they incorporated findings from the MaxART study in the revised manuscript, but I did not see any manuscripts from MaxART in the references. I think that the authors could do a better job linking this manuscript to more recent studies and reviews of treatment as prevention.

In addition, I disagree with the description of the study as "mixed methods", as no qualitative data was collected or analyzed. The interviews appeared to have been structured and quantitative. If qualitative data was collected, then please include details on the qualitative data collection methods (including interview guide as an appendix), and qualitative analysis methods (grounded theory, framework, other approaches to coding and analysis).

Finally, in responding to reviewers comments, it would be helpful to provide more details, such as the page and line numbers, where the changes were made.

7. PLOS authors have the option to publish the peer review history of their article (what does this mean?). If published, this will include your full peer review and any attached files.

Reviewer #1: **Yes: **Yunia Mayanja

Reviewer #2: No

---

## [Author Response · Author response to Decision Letter 1]

15 Nov 2020

Please refer to our response memo.

---

## [Editor Report · Decision Letter 2]

26 Nov 2020

Effects of the implementation of the HIV Treat All guidelines on key ART treatment outcomes in Namibia

PONE-D-20-12826R2

Dear Dr. Vu,

We’re pleased to inform you that your manuscript has been judged scientifically suitable for publication and will be formally accepted for publication once it meets all outstanding technical requirements.

Kind regards,

Matthew Quaife

Academic Editor

PLOS ONE
---

## [Editor Report · Acceptance letter]

9 Dec 2020

PONE-D-20-12826R2 

Effects of the implementation of the HIV Treat All guidelines on key ART treatment outcomes in Namibia 

Dear Dr. Vu:

I'm pleased to inform you that your manuscript has been deemed suitable for publication in PLOS ONE. Congratulations! Your manuscript is now with our production department. 

Kind regards, 

on behalf of

Dr. Matthew Quaife 

Academic Editor

PLOS ONE